# Knowledge Network Embedding of Transcriptomic Data from Spaceflown Mice Uncovers Signs and Symptoms Associated with Terrestrial Diseases

**DOI:** 10.3390/life11010042

**Published:** 2021-01-12

**Authors:** Charlotte A. Nelson, Ana Uriarte Acuna, Amber M. Paul, Ryan T. Scott, Atul J. Butte, Egle Cekanaviciute, Sergio E. Baranzini, Sylvain V. Costes

**Affiliations:** 1Integrated Program in Quantitative Biology, University of California San Francisco, San Francisco, CA 94143, USA; Charlotte.Nelson@ucsf.edu; 2Space Biosciences Division, NASA Ames Research Center, Moffett Field, CA 94035, USA; ana.e.uriarteacuna@nasa.gov (A.U.A.); amber.m.paul@nasa.gov (A.M.P.); ryan.t.scott@nasa.gov (R.T.S.); egle.cekanaviciute@nasa.gov (E.C.); 3KBR, NASA Ames Research Center, Moffett Field, CA 94035, USA; 4NASA Postdoctoral Program, Universities Space Research Association (USRA), Mountain View, CA 94043, USA; 5Bakar Computational Health Sciences Institute, University of California San Francisco, San Francisco, CA 94143, USA; atul.butte@ucsf.edu; 6Department of Pediatrics, University of California San Francisco, San Francisco, CA 94143, USA; 7Weill Institute for Neuroscience, Department of Neurology, University of California San Francisco, San Francisco, CA 94143, USA

**Keywords:** spaceflight, knowledge graph, transcriptomics

## Abstract

There has long been an interest in understanding how the hazards from spaceflight may trigger or exacerbate human diseases. With the goal of advancing our knowledge on physiological changes during space travel, NASA GeneLab provides an open-source repository of multi-omics data from real and simulated spaceflight studies. Alone, this data enables identification of biological changes during spaceflight, but cannot infer how that may impact an astronaut at the phenotypic level. To bridge this gap, Scalable Precision Medicine Oriented Knowledge Engine (SPOKE), a heterogeneous knowledge graph connecting biological and clinical data from over 30 databases, was used in combination with GeneLab transcriptomic data from six studies. This integration identified critical symptoms and physiological changes incurred during spaceflight.

## 1. Introduction

NASA recognizes five main hazards of spaceflight to human health, including altered gravity (microgravity and hypergravity), ionizing radiation, isolation/confinement, hostile/closed environment, and distance from Earth. These health risks caused by the space environment resemble multiple disorders found on Earth, including muscle atrophy and bone loss, cardiovascular deconditioning, immune dysfunction, and central nervous system deficits [1]. Therefore, repurposing current FDA-approved treatments for issues that arise during spaceflight could significantly reduce the time needed to develop new therapeutics and limit their side effects.

Since its establishment in 2015, NASA GeneLab [2] has become a prominent open-source repository of data from real and simulated spaceflight studies. This platform has enabled computational analysis of multi-omics data, visualization of results, and integration with descriptive metadata, such as environmental data (e.g., space radiation dosimetry). GeneLab has already supported dozens of published studies, created a global collaboration to develop uniform standards for spaceflight-omics [3], and resulted in new space biology discoveries [4,5]. However, it has not yet been possible to use NASA GeneLab to combine and compare space and terrestrial data. Such capability would be a major advancement in fundamental spaceflight biology and its applications, including identifying new targets or repurposing terrestrial therapeutics for spaceflight countermeasures.

NASA GeneLab is planning to set up a portal dedicated to computational modeling that enables comparisons between datasets in addition to already existing data input, query, analysis, and visualization capabilities. Knowledge graphs (KGs) would be a suitable approach to facilitate this goal by unifying disparate datasets into a human queryable framework. KGs have already been widely adopted in biomedical research to unravel the complex relationship between biological changes and disease phenotypes [6,7,8,9,10].

Specifically, a new massive UCSF-based KG database, the Scalable Precision Medicine Oriented Knowledge Engine (SPOKE) has transformed structured data from over 30 human biomedical databases (-omics, chemical structures, molecular and cellular responses, physiological data including e.g., patient symptoms and drug side effects, etc.) into a KG with almost 400,000 nodes of 12 types and over 10 million edges of 32 types [11,12]. Therefore, SPOKE has the potential to be combined with the NASA GeneLab modeling portal, expanding it to link terrestrial biomedical sciences to space biosciences research and space medicine.

In this study, we integrated data from six different NASA GeneLab datasets in SPOKE to enable normalization that highlighted new nodes defining systems and effects that are known to be relevant for space travel but would have been impossible to uncover without using SPOKE (workflow Figure 1a). These results suggest that SPOKE can be utilized to gain a deeper biological understanding of the health hazards associated with spaceflight and provide the proof of concept for its broader utilization to integrate space and terrestrial biological data.

## 2. Materials and Methods

### 2.1. GeneLab Data Processing and Analysis

Gene expression data were downloaded from the NASA GeneLab repository (https://genelab-data.ndc.nasa.gov/), datasets GLDS-4, GLDS-244, GLDS-245, GLDS-246, GLDS-288, and GLDS-289. All data had been processed and analyzed using standard NASA GeneLab techniques detailed below. Matched flight/live animal return versus ground control data was used for analysis.

Previously, raw data were processed separately for each dataset by the NASA GeneLab data processing team. For datasets containing RNA Sequencing (RNA-Seq) assays (GLDS-244, GLDS-245, GLDS-246, GLDS-288, GLDS-289), raw FASTQ files were assessed for the percentage of rRNA using HTStream SeqScreener (version 1.1.0 for GLDS-244, GLDS-245, GLDS-246 and version 1.3.1 for GLDS-288, GLDS-289; https://s4hts.github.io/HTStream/) and filtered using Trim Galore! (version 0.6.4; https://www.bioinformatics.babraham.ac.uk/projects/trim_galore/) [13]. Raw and trimmed fastq file quality was evaluated with FastQC [14] (version 0.11.9). MultiQC [15] (version 1.8 for GLDS-244, GLDS-245, GLDS-246 and version 1.9 for GLDS-288, GLDS-289) was used to generate MultiQC reports. Mus musculus STAR [16] and RSEM [17] references were built using STAR (version 2.7.1a for GLDS-244, GLDS-245, GLDS-246 and version 2.7.4a for GLDS-288, GLDS-289) and RSEM (version 1.3.1), respectively, genome version mm10-GRCm38 (Mus_musculus.GRCm38.dna.toplevel.fa), and the following gtf annotation file: Mus_musculus.GRCm38.96.gtf. Trimmed reads were aligned to the Mus musculus STAR reference with STAR (version 2.7.3a for GLDS-244, GLDS-245, GLDS-246 and version 2.7.4a for GLDS-288, GLDS-289) and aligned reads were quantified using RSEM (version 1.3.1 from the NASA GeneLab repository).

Data representing the quantitative analysis of gene expression for each dataset was downloaded from the NASA GeneLab repository, where it had been previously analyzed and imported into R [18] (version 3.6.3). It was then combined to create a gene counts table containing the data for all samples of every dataset. For GLDS-244, GLDS-245, and GLDS-246, only non-ERCC (External RNA Controls Consortium [19], i.e., a spike-in mixture used for normalization) genes were used. Data was normalized with DESeq2 [20] (version 1.26.0). A principal component analysis was performed using prcomp (stats version 3.6.3) and plotted using plotly [21] (version 4.9.2.1). For datasets containing DNA microarray assays (GLDS-4) raw CEL files were read in and normalized using an in-house R script as described [22].

To quantify overlapping pathways between GLDS-244, -245, and -246, Entrez Gene IDs of genes that showed a significant difference (*p* < 0.05) between 29-day flight/live animal return and ground controls were used as the input to Molecular Signatures Database v7.2, GeneOntology [23,24,25] (GO) gene sets. (GO biological process, GO cellular component, GO molecular function). The top 50 statistically significant gene sets were compared to identify overlaps. The same approach was applied to quantify the overlapping gene sets between GLDS-288 and -289.

### 2.2. Scalable Precision Medicine Oriented Knowledge Engine

Scalable Precision Medicine Oriented Knowledge Engine (SPOKE) [11,12] is a population level heterogeneous knowledge graph. SPOKE was generated by unifying over 30 publicly available databases. Currently, SPOKE contains almost 400,000 nodes of 12 types (*Anatomy*, *BiologicalProcess*, *CellularComponent*, *Compound*, *Disease*, *Gene*, *MolecularFunction*, *Pathway*, *PharmacologicalClass*, *Protein*, *SideEffect*, and *Symptom*). These nodes are connected by 32 types of biologically meaningful edges (*n* > 10 million).

### 2.3. Gene-Specific Propagated SPOKE Entry Vectors

Propagated SPOKE Entry Vectors (PSEVs) are generated using a modified version of topic-specific page rank to learn and embed the importance of each node in SPOKE for a given restart node or set of nodes [26,27]. These restart nodes, called SPOKE Entry Points (SEPs), are any concept in the input data that overlaps with a node(s) in SPOKE [28]. In this analysis, the SEPs were the mouse genes that have homologs to the human *Gene* nodes in SPOKE. A Gene PSEV was produced by allowing a random walker to traverse the edges in SPOKE and then forcing them to restart at a specific *Gene* SEP. The forced restart ensures that the walker will spend the majority of time on nodes that are important for that *Gene*. The significance of each node is then stored in an element of the PSEV such that the length of the PSEV is equal to the number of nodes in SPOKE (*n* = 389,297). Code used to generate the data in this manuscript is available at (https://doi.org/10.5281/zenodo.4408540).

### 2.4. Integrating Gene Expression Data and PSEVs

For each study, the −log_2_ fold-change (FC) mouse gene expression data was mapped to the human gene nodes in SPOKE. The homologous mapping between species was achieved using HomoloGene IDs [29]. If multiple mouse genes mapped to a single human gene, then the average FC was used. Additionally, some studies contained multiple comparisons between space and ground or baseline control mice. An example of this is the study GLDS-244 that compared mice at two space-time points (day-29 and days 53–56). In these instances, genes were removed if the FC comparisons were not in the same direction (i.e., if space versus ground day-29 had a positive FC and days-53–56 had a negative FC). This filter focuses on the data set of genes that remain consistent during space travel.

After genes were mapped and filtered for a given study, the pre-computed PSEVs for the remaining genes were extracted. This PSEV matrix was z-score normalized and then ranked such that the most important node in a given PSEV was equal to the number of nodes in SPOKE (*n* = 389,297) and the least important was ranked one. Then for each comparison, the filtered PSEV matrix was adjusted using the FCs. This was accomplished by taking the product of a single column in the FC matrix and the filtered normalized PSEV matrix. It is necessary for the rows (genes) of the filtered normalized PSEV matrix to be in the same order as the rows in the FC matrix. Next, each column (node) in the adjusted PSEV-matrix was summed resulting in a vector in which each element or position corresponded to a node in SPOKE (length = 389,297). Each node was then ranked as before (with the highest value in the vector ranked 389,297). In practice, this was achieved by taking the dot product of the filtered FC matrix (transposed) and the filtered normalized PSEV matrix and then ranking the resulting matrix.

### 2.5. Finding Significant Spoke Nodes

The PSEV comparisons from the six studies were pooled together and separated into three groups (Ground vs. Baseline, Space vs. Baseline, and Space vs. Ground). Welch’s *t*-test was used to evaluate whether the distribution of ranks of a given node in the Ground vs. Baseline group was significantly different from that in either Space vs. Baseline or Space vs. Ground (Appendix A). Top nodes, those that were ranked significantly different in either space travel comparisons (Space vs. Baseline and Space vs. Ground) than in Ground vs. Baseline, were identified using the *p*-values from the Welch’s *t*-test. Since 159,374 nodes had a *p*-value < 0.025 in either or both space travel comparisons, top nodes were further filtered by selecting the most significant 2.5% of each node type for Space vs. Ground and/or Space vs. Baseline (*n* = 15,801; 4.1%).

### 2.6. Retracing Paths from Input Gene to SPOKE Node

A high correlation between a gene’s FC and the rank of a specific node suggests that the gene FC is at least partially responsible for the prioritization of the node within a PSEVs. The correlation was calculated between genes (present in >20% of FC comparisons; *n* = 7567) and a set of top *Anatomy*, *BiologicalProcess*, *CellularComponent*, *MolecularFunction*, *Pathway*, and *Symptom* nodes (*n* = 22). Next, paths were found between genes that had a high correlation (correlation > 0.6) and the set of top nodes. Gene-node pairs were then filtered to only include pairs that had the same sign (positive gene expression and positive Welch *t*-statistic). Then, in order to visualize paths between gene-node pairs, paths were filtered to have a maximum of three edges and less than 100 possible combinations of nodes within the path. This left over 17,000 gene-node pairs and 234,000 possible paths.

The paths shown were selected based on their simplicity and the FC of the original genes (Appendix A). The *p*-values, derived when calculating the FCs used as input for PSEV creation, were combined for Ground vs. Baseline and the space travel groups (Space vs. Baseline and Space vs. Ground together) using Stouffer’s method [30]. Each gene FC was judged on whether the average space travel group had a combined *p*-value that was more significant than Ground v Baseline (Appendix A, y-axis). Then the Welch’s *t*-test was used to determine whether the FC distributions were significantly different between groups. Space vs. Baseline and Space vs. Ground distributions were compared to the Ground vs. Baseline separately and then averaged (Appendix A, x-axis).

## 3. Results

### 3.1. Transcriptional Profiling of Mice after Space Flight

Here we conducted a meta-analysis of six independent transcriptomic datasets (GLDS-4, -244, -245, -246, -288, and -289) from experimental mice obtained during four different spaceflight missions (STS-118, TCU (SpaceX-9), MHU-2 (Space X-12), and RR-6 (SpaceX-13)), at five time- points of collection (13-, 29-, 30-, and 35-days live animal return (LAR); and 53–56 days (ISS terminal)), on the International Space Station (ISS) (Figure 1b and Table 1). While experiments varied in their design (i.e., duration of flight, age at launch, the genotype of mice, transcriptomic platform, time of collection), the objective of these experiments was to identify changes in gene expression induced by spaceflight in three different immune-related organs—thymus (primary lymphoid organ), spleen (secondary lymphoid organ), and liver (immune-associated/digestive organ, with lymphatic cells playing a role in its responses to injury [31]).

These sample sets were selected to include multiple immune-associated organs (thymus, spleen, liver) collected from the same space-flown mice as well as between mice flown on different missions to increase sample diversity and to include RNA sequencing and microarray as two different sequencing methods to show that both can be used as inputs to SPOKE.

After data normalization, the principal component analysis revealed a strong separation of samples by mission and tissues (Figure 2a). These findings are unsurprising, given that these variables are confounding factors of different missions/collections. However, we also observed that samples from the same time point of mission/collection from two different experiments clustered together, suggesting that some biological effects were captured. When PCA was used to plot samples from similar experimental conditions (space-flown, ground, and baseline from the same RR-6 mission), no obvious separation between samples obtained during flight, baseline, and the ground was observed (Figure 2b).

Differentially expressed genes were identified in the thymus, liver, and spleen in space-flown mice vs. ground controls after live animal return from the RR-6 (SpaceX-13) mission. Furthermore, using the differentially expressed genes as an input to pathway analysis (by a hypergeometric test) showed a number of statistically significant biological functions dysregulated by space flight in the thymus, liver, and spleen, including some that overlapped between the tissues (Figure 2c). While some gene sets were tissue-specific, nine of them were shared among the three tissues, including apoptosis, cell metabolic process, and cell membrane integrity (Figure 2d).

### 3.2. Fold-Change Enhanced Propagated SPOKE Entry Vectors

While established methods of transcriptional profiling can inform about dysregulated molecular pathways, they provide little insight into higher-order phenotypes, such as associated signs and symptoms of disease. Using SPOKE, a KG that integrates information of both biological and clinical databases, it is possible to score every node of the graph as a function of the “information flow” elicited by a defined set of quantitative inputs. SPOKE leverages the complexity of the hierarchical organization of complex organisms to identify nodes with shared information flow (regardless of whether the input itself was significant or not).

Gene-specific Propagated SPOKE Entry Vectors (PSEVs) were generated from the selected GeneLab studies prior to integrating gene expression results with SPOKE [11,12]. Each gene-specific PSEV was created using a modified version of topic-specific page rank [26,27] in which the random walker was forced to restart at the corresponding *Gene* node in SPOKE (See Methods, Figure 3a). This focused the random walker on nodes that were the most important for a given node (in this case, *Gene* node since the input is gene expression). The amount of time a random walker spent on a node was then stored in a defined element (position within) of the PSEV vector. All PSEVs were then stored in the pre-computed PSEV matrix. For each gene expression study, the pre-computed PSEV matrix was filtered and normalized to match the genes within the study (Figure 3b; Methods). The dot product was then used with the normalized PSEV matrix and the −log_2_ fold-change (FC) to produce the PSEVs for that study. After PSEVs were computed for each study, they were pooled and separated into specific experimental groups to enable meaningful comparisons to test the hypothesis that spaceflight alters gene expression (Ground vs. Baseline, Space vs. Baseline, and Space vs. Ground) (Figure 3c).

Each element in a PSEV corresponds to a single node in SPOKE. Therefore, it is possible to determine the overall significance of a node for spaceflight by evaluating the differential distribution of node ranks in the PSEV. Welch’s *t*-test [32] was utilized to compare a node’s rank distribution in the Ground vs. Baseline to that in either Space vs. Baseline or Space vs. Ground (Appendix A).

Strikingly, nodes that are known to be relevant for space travel such as space motion sickness (*Symptom*), regulation of blood vessel diameter (*BiologicalProcess*), taste receptor complex (*CellularComponent*), Vitamin D (calciferol) metabolism (*Pathway*), and sympathetic nervous system (*Anatomy*) scored among the top 5% of nodes (top 2.5% per type for Space vs. Baseline and/or Space vs. Ground). Figure 4 shows violin plots from a select set of nodes (*n* = 22) in SPOKE that had significantly different ranks in spaceflight (Space vs. Baseline and/or Space vs. Ground) compared to Ground vs. Baseline. From these, 11 correspond to symptoms (pink boxed violin charts, Figure 4a), five to gene ontology/pathway concepts (teal boxed violin charts, Figure 4b–d), and six to anatomical regions (green boxed violin charts, Figure 4e). Violin plots for each category, sub-networks demonstrate how the gene expression results drive information from these 22 nodes. Among the other biological top nodes were nodes that reflected the results of the original studies such as those related to t-cell activity, regulation of stress, and TGFβ1 [1,33].

Taken together, these results show that potential human physiological changes during spaceflight can be inferred by embedding mouse gene expression data with a KG that integrates observed concepts (i.e., genes) with unobserved, higher-order phenotypes associated with each other in a biologically meaningful manner.

## 4. Discussion

One of the major objectives of biomedical research is to advance our understanding of human diseases in order to develop effective countermeasures. This aim becomes considerably more challenging when the physiological changes arise from spaceflight. Major efforts have been made by NASA GeneLab to collect and provide multi-omics data from model organisms. Additionally, NASA GeneLab data brought into the SPOKE system could be complemented by including murine phenotypical pathophysiological and biochemical non-omics data (more nodes) from the Ames Life Sciences Data Archive [34], and eventually the SPOKE system could be used for human spaceflight research data related to astronauts. However, the major challenges of analyzing any datasets generated during spaceflight are their low statistical power, considerable heterogeneity, and limited reproducibility [35]. These limitations are largely accepted by the scientific community as a reasonable trade-off for the novelty and potential for discovery these experiments entail. As a new strategy to maximize the utility of these datasets, we propose the data from model organisms can be integrated through a knowledge graph (KG) such as SPOKE. KGs including SPOKE, are bounded by present day biological knowledge. As a result, inferences made through SPOKE may change as our biological data and knowledge expands.

Here, we report the results of a KG-driven, meta-analysis of six murine transcriptomic studies (five RNAseq and one microarray) from NASA GeneLab. The samples were taken from three distinct anatomical sites (thymus, liver, and spleen) and covered multiple spaceflight duration and gravity conditions. PCAs using only gene expression data illustrated that most of the differences between the samples could be attributed to either the study or the anatomical site.

Next, we hypothesized that, though this data came from a diverse set of experiments, SPOKE embeddings (i.e., “signatures”) could be used to recover space travel changes that are conserved across the studies. To accomplish this, −log_2_ fold-change gene expression (FC) data from each study was applied to gene-specific Propagated SPOKE Entry Vectors (PSEVs). Gene-specific PSEVs are vectors that describe how important each node in SPOKE is for a given gene. Therefore, multiplying PSEVs by FC data will highlight nodes that are both important for the input gene set and to prioritize them according to how differentially expressed the input genes are.

PSEVs from all of the studies were then pooled together and separated into three groups based on the type of FC comparison (Ground vs. Baseline, Space vs. Baseline, and Space vs. Ground). The distribution of node rank was analyzed for each node and the top 5% were selected for each node type. These top nodes were enriched for nodes for phenotypes and physiological changes known to be impacted by spaceflight. Furthermore, paths were found between the input gene set and the top node set. These paths shed light onto the underpinnings of spaceflight related health hazards and could potentially be used to identify drug targets. In the future, archived spaceflight and other experimental samples could be used to validate the predicted signatures and assess their physiological significance without the need for further experiments. Thus, we anticipate that our results are the very first steps towards a broader collaboration utilizing the SPOKE model to compare spaceflight and terrestrial phenotypes.

There is increasing interest in developing personalized risk predictions and treatments in support of long-duration deep space missions [36]. Thus, expanding the computational approaches from the *general* comparison of spaceflight and terrestrial diseases to using input from a single subject to map their *individual* risk profile would allow developing optimal medical care for individual astronauts. Notably, the power of SPOKE stems from a wide variety of its inputs that combine multi-omics, clinical, and physiological data, which may provide a useful complement to the currently utilized risk management tools that are based upon probabilistic mathematical modeling and simulations [37].

Using a knowledge graph connecting molecular and physiological entities (among others) via biologically relevant relationships constitutes a significant advancement for complex, heterogeneous data analysis. This approach complements conventional transcriptomics analysis by extending the biological significance to higher-level phenotypes such as symptoms and side effects, which is not possible with current methods. In the long-term perspective, the SPOKE platform may also be of value to mission planners such as the NASA Human Systems Risk Board.

## Figures and Tables

**Figure 1 life-11-00042-f001:**
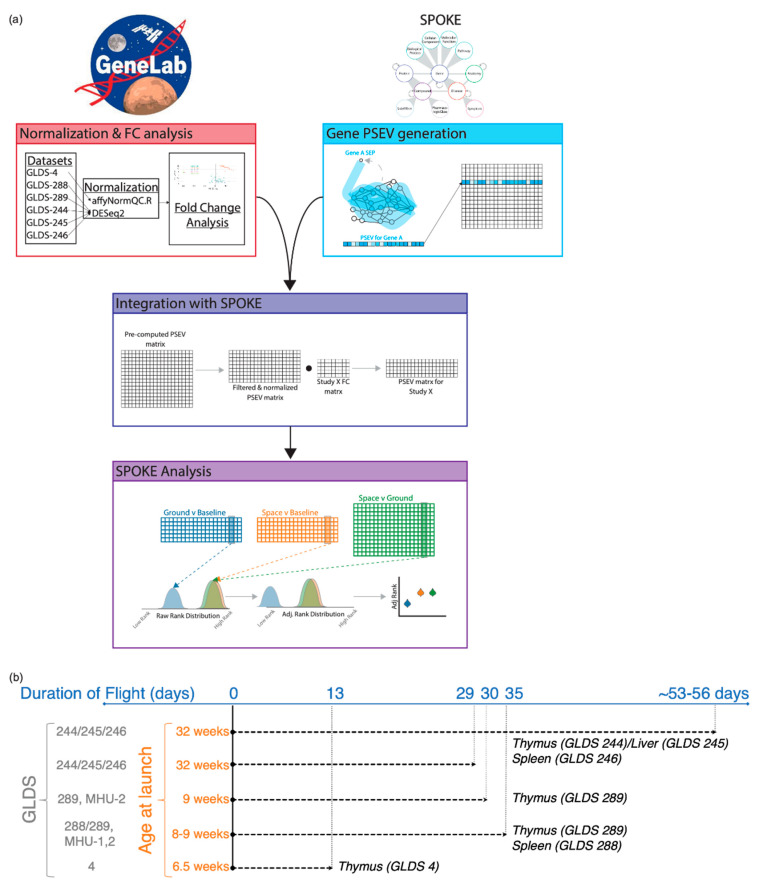
Analysis workflow and summary of experimental conditions across GeneLab datasets used for the analysis. (**a**) Workflow depicting the different stages of the analysis. (**b**) Datasets GLDS-4, -244, -245, and -246 used C57BL/6NTac mice. Datasets GLDS-288 and -289 used C57BL/6J mice for spaceflight and both C57BL/6J and Charles River mice for ground controls.

**Figure 2 life-11-00042-f002:**
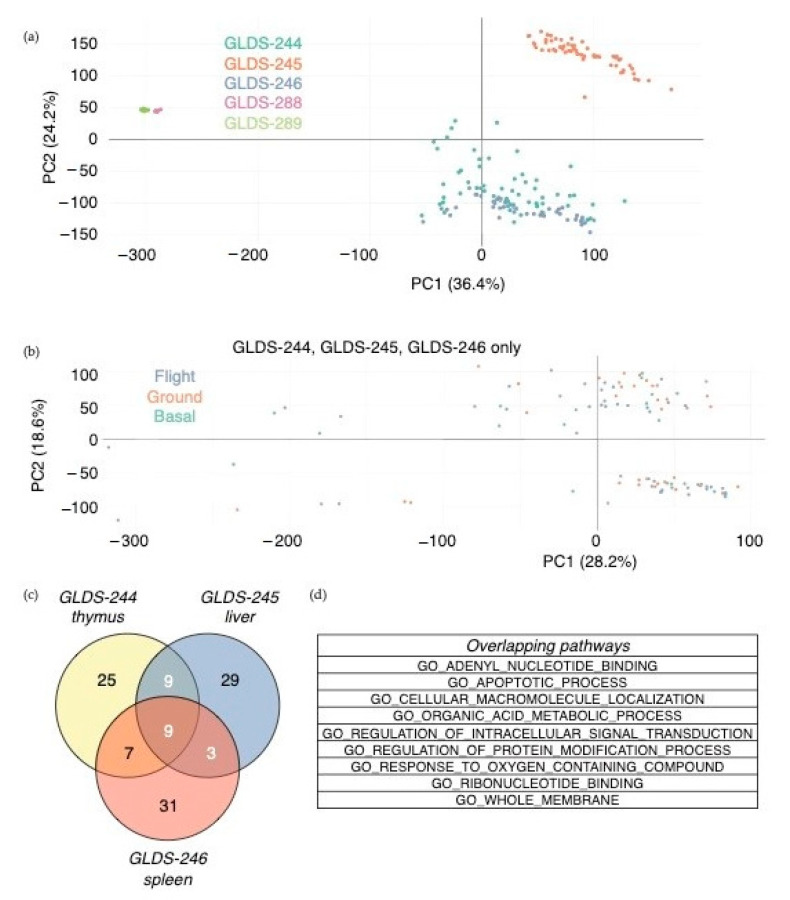
Transcriptomic analysis of spaceflight-associated changes in gene expression. (**a**) Principal component analysis of all samples, colored by the dataset. (**b**) Principal component analysis of datasets GLDS-244, -245, and -246, colored by flight condition. (**c**,**d**) Overlapping gene sets between datasets GLDS-244, 245, and 246 out of the top 50 Gene Ontology gene sets using significantly differently expressed genes (*p* < 0.05) between flight and ground conditions, live animal return after 29 days on the ISS. Venn diagram showing overlapping gene sets between datasets (**c**) and the list of gene sets overlapping between all three datasets (**d**). Three out of the top 50 gene ontology (GO) gene sets overlapped between datasets GLDS-288 and -289, none of which overlapped with GLDS-244, -245, and -246.

**Figure 3 life-11-00042-f003:**
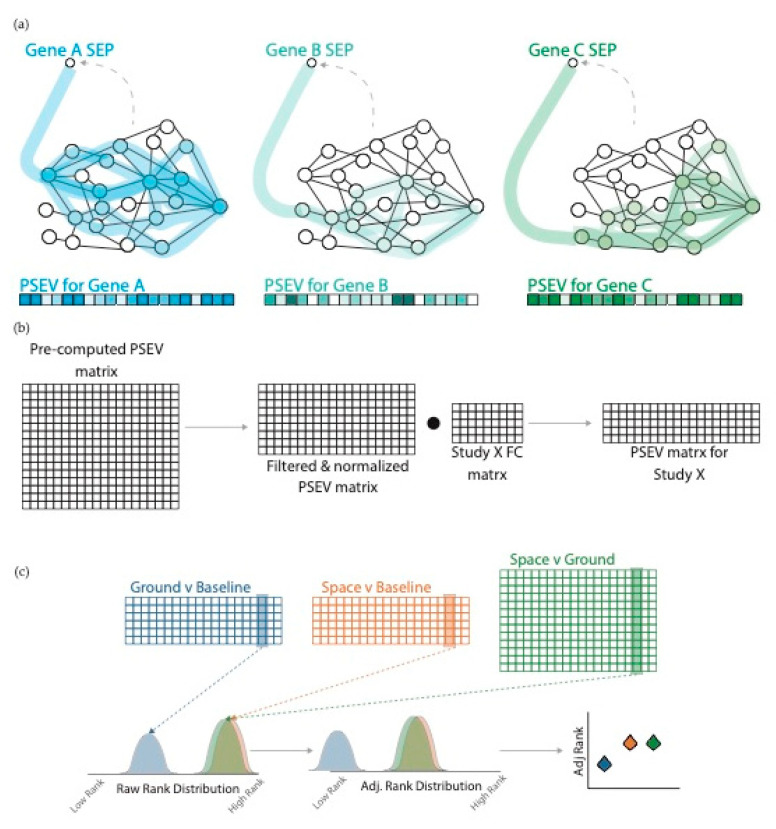
Propagated SPOKE Entry Vectors (PSEVs) using gene expression fold-change (FC). (**a**) PSEVs were pre-computed for all SPOKE genes. For each gene, the random walker was forced to restart at that gene (probability of random jump = 0.1). After PSEVs were finished they were stored in the pre-computed PSEV matrix. (**b**) For each study, the pre-computed PSEV matrix was filtered and normalized. Then the dot product was taken between the normalized matrix and the FC matrix to generate the PSEV matrix for that study. (**c top**) The PSEV matrices for each study were pooled together and separated into groups: Ground vs. Baseline (blue), Space vs. Baseline (yellow), and Space vs. Ground (green). (**c bottom**) The distributions of the node ranks were adjusted using the mean Ground vs. Baseline rank.

**Figure 4 life-11-00042-f004:**
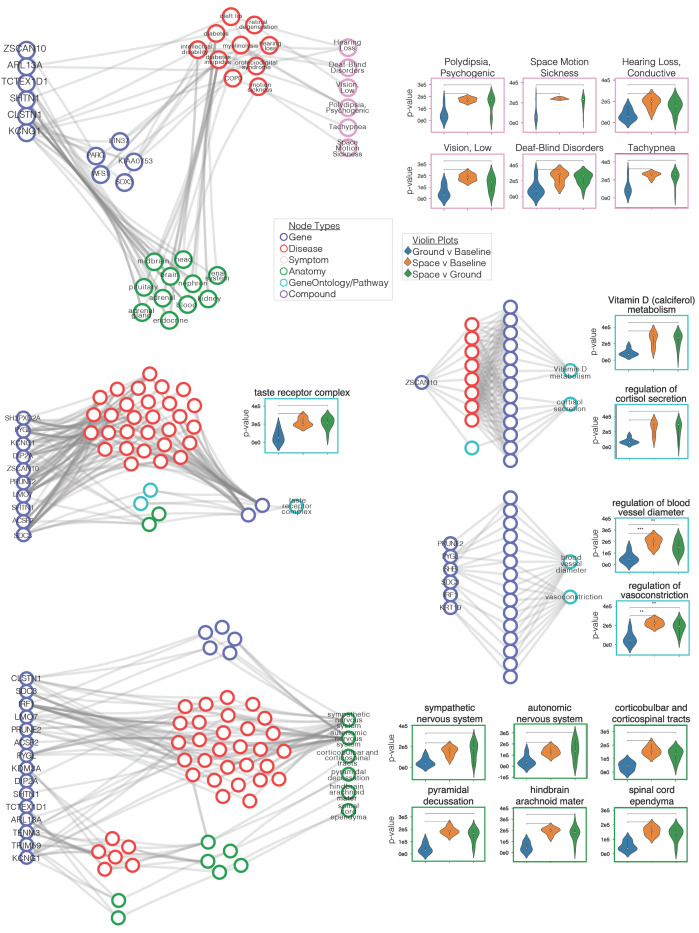
Retracing paths between genes and top nodes. Gene expression FC values drive information flow to nodes in SPOKE. (**a**–**e**) Paths were traced between genes that were partially responsible for pushing information to a set of significant nodes (*n* = 22). These paths were shown for (**a**) 10 *Symptom* nodes, (**b**) *taste receptor complex* (*CellularComponent*), (**c**) *regulation of cortisol secretion* (*BiologicalProcess*) and *Vitamin D* (*calciferol*) *metabolism* (*Pathway*), (**d**) *regulation of vasoconstriction* (*BiologicalProcess*) and *regulation of blood vessel diameter* (*BiologicalProcess*), and (**e**) six *Anatomy* nodes. Violin plots for each significant node show that the ranks within Space vs. Baseline and/or Space vs. Ground separated from the Ground vs. Baseline. In each violin plot Ground vs. Baseline (blue), Space vs. Baseline (yellow), and Space vs. Ground (green).

**Table 1 life-11-00042-t001:** Descriptive metadata for each NASA GLDS dataset analyzed by the Scalable Precision Medicine Oriented Knowledge Engine (SPOKE).

	**GeneLab Study**	**Tissue**	**Sequencing Type**	**Strain**		**Mission/Flight**	**Flight Duration**
**A.**	GLDS-4	Thymus	Microarray	C57BL/6NTac		STS-118	13-days (12.76 day)
	GLDS-244	Thymus	RNA-sequencing	C57BL/6NTac		RR-6 (SpaceX-13)	29-days (*n* = 9, LAR); 53–56-days (*n* = 10, ISS terminal)
	GLDS-245	Liver	RNA-sequencing	C57BL/6NTac		RR-6 (SpaceX-13)	29-days (*n* = 9, LAR); 53–56-days (*n* = 10, ISS terminal)
	GLDS-246	Spleen	RNA-sequencing	C57BL/6NTac		RR-6 (SpaceX-13)	29-days (*n* = 9, LAR); 53–56-days (*n* = 10, ISS terminal)
	GLDS-288	Spleen	RNA-sequencing	C57BL/6J (flight); Charles River Laboratories Japan (GC)		TCU (SpaceX-9)	35-days
	GLDS-289	Thymus	RNA-sequencing	C57BL/6J (flight); Charles River Laboratories Japan (GC)		TCU (SpaceX-9, MHU-1; SpaceX-12, MHU-2)	35-days MHU-1; 30-days MHU-2
	**GeneLab Study**	**Age at Initiation**	**Age at Euthanasia**	**Sex**	**Sample Size (n/Cohort)**	**Controls**	**Collection Location**
**B.**	GLDS-4	~6.5-weeks	8-weeks	n/a	FLT (*n* = 4); GC (*n* = 4)	Synchronous Ground Controls (GC)	Ground post-flight
	GLDS-244	32-weeks	36-weeks LAR; 44-weeks ISS terminal; 36-weeks LAR/ISS terminal Baseline GC; 41-weeks LAR GC; 44-weeks ISS Terminal GC	Female	LAR (*n* = 9); ISS terminal (*n* = 10); Baseline LAR (*n* = 10); Baseline ISS Terminal (*n* = 9); LAR GC (*n* = 9); ISS Terminal GC (*n* = 10)	Baseline (LAR, ISS terminal); synchronous GC (LAR, ISS terminal)	4-days post-flight (LAR); 53–56-day In-flight (ISS terminal)
	GLDS-245	32-weeks	36-weeks LAR; 44-weeks ISS terminal; 36-weeks LAR/ISS terminal Baseline GC; 41-weeks LAR GC; 44-weeks ISS Terminal GC	Female	LAR (*n* = 9); ISS terminal (*n* = 10); Baseline LAR (*n* = 10); Baseline ISS Terminal (*n* = 9); LAR GC (*n* = 9); ISS Terminal GC (*n* = 10)	Baseline (LAR, ISS terminal); synchronous GC (LAR, ISS terminal)	4-days post-flight (LAR); 53–56-day In-flight (ISS terminal)
	GLDS-246	32-weeks	36-weeks LAR; 44-weeks ISS terminal; 36-weeks LAR/ISS terminal Baseline GC; 41-weeks LAR GC; 44-weeks ISS Terminal GC	Female	LAR (*n* = 9); ISS terminal (*n* = 10); Baseline LAR (*n* = 10); Baseline ISS Terminal (*n* = 9); LAR GC (*n* = 9); ISS Terminal GC (*n* = 10)	Baseline (LAR, ISS terminal); synchronous GC (LAR, ISS terminal)	4-days post-flight (LAR); 53–56-day In-flight (ISS terminal)
	GLDS-288	8-weeks	12-weeks	Male	Spaceflight (MG, *n* = 3); Spaceflight w/centrifugation (AG, *n* = 3); Synchronous (GC, *n* = 3)	Spaceflight w/centrifugation; Synchronous GC	Ground post-flight
	GLDS-289	8-weeks MHU-1; 9-weeks MHU-2	12-weeks	Male	Spaceflight (MG, MHU-1, *n* = 3); Spaceflight w/centrifugation (AG, MHU-1, *n* = 3); Synchronous (GC, MHU-1, *n* = 3); Spaceflight (MG, MHU-2, *n* = 3); Spaceflight w/centrifugation (AG, MHU-2, *n* = 3); Synchronous (GC, MHU-2, *n* = 3)	Spaceflight w/centrifugation (AG, MHU-1); Synchronous (GC, MHU-1); Spaceflight w/centrifugation (AG, MHU-2); Synchronous (GC, MHU-2)	Ground post-flight

“GC” denotes ground control, “RR” denotes rodent research, “TCU” denotes transportation case units.

## Data Availability

Not applicable.

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
