# Peer review of "Knowledge Network Embedding of Transcriptomic Data from Spaceflown Mice Uncovers Signs and Symptoms Associated with Terrestrial Diseases"

_life, 2021, doi:10.3390/life11010042_

Round 1
Reviewer 1 Report
In the paper the author presents an application to explore 6 expression dateset of mouse spaceflight studies by exploiting a human Knowledge Graphs (KG) called SPOKE. They propose to integrate the log fold changes of the different comparisons present in the datasets they analyzed with KG to prioritize the KG nodes that are mostly related to known pato-physiological changes that arise during spaceflight.
Even if there is a limited overall methodological novelty, the application is interesting because allows to connect molecular changes to phenotype changes. Nevertheless, some method aspectecs and some parameters decisions could have been better explained i.e. the choice of the top 5% genes rather that p-values (see specific comments).
Specific comments
Page 4 134 - 139 The method explained was very hard to follow. The paper will benefit a rewriting of this portion. For the reader, it could be interesting to know the rationale behind taking the dot product between FC matrix and PSEV matrix.
Page4 line 145-146. “Top nodes were selected using the most significant 2.5% per node type for Space vs. Ground and/or Space vs. Baseline (n=15,875; 4.1%)” can the author provide an explanation why they used the 2.5% cut of rather p-value cut-off. The significant nodes were too few or too much?
Page 4 line 150: please clarify the meaning of top nodes.
Page 4 line 159 The authors wrote: “The p-values of the FCs used as input for PSEV creation were averaged for each group (Ground vs. Baseline, Space vs. Baseline, and Space vs. Ground). This gives us an estimate of how significant the gene FC was for a group as a whole.” This sentence is not correct from a statistical point of view: the average of p values is not an estimate of the significance of the gene in the whole group. Also the sentence starting at line 161 is not statistically sound (especially “that on average was more significant”). If you use arbitrary p-values combination to build a score you should state that the statistical meaning is lost. Otherwise use a p-value combination method that is statistically valid.
Minors
Page 5 line 199 Figure 2C does not show gene overlap. Moreover it is not clear if “these genes” refers to DEGs or to the overlap.
Page3 Line 105 correct G in GO
Page3 line 124 please clarify the meaning of the numbers 389 , 297 in “PSEV is equal to the number of nodes in SPOKE (n = 389 , 297)”
Page 4 line 146: explain the meaning of 'n' and the percentage
Page5 Line 184 If PMID:27965673 was meant as a reference, it is not immediately clear to me the link between the topic of the paper.
Line 197 - 203 Figure 2 c,d caption does not match the figure and the text. Please clarify the figure caption or the text.
Table 1 at page 6 is not readable. Please split long content in multiple lines.
Figure 4 is not readable as well
Author Response
Reviewer #1
In the paper the author presents an application to explore 6 expression dateset of mouse spaceflight studies by exploiting a human Knowledge Graphs (KG) called SPOKE. They propose to integrate the log fold changes of the different comparisons present in the datasets they analyzed with KG to prioritize the KG nodes that are mostly related to known pato-physiological changes that arise during spaceflight.
Even if there is a limited overall methodological novelty, the application is interesting because allows to connect molecular changes to phenotype changes. Nevertheless, some method aspectecs and some parameters decisions could have been better explained i.e. the choice of the top 5% genes rather that p-values (see specific comments).
Specific comments
Page 4 134 - 139 The method explained was very hard to follow. The paper will benefit a rewriting of this portion. For the reader, it could be interesting to know the rationale behind taking the dot product between FC matrix and PSEV matrix.
The process to embed gene expression changes (FC) onto SPOKE involves taking a single column in the FC matrix (each row corresponds to a single gene) and multiplying that with the pre-computed PSEV matrix (such that the gene -rows- in the PSEV matrix directly correspond with the rows in the FC column. With the resulting matrix we then sum each column (nodes in spoke). This results in a vector with a length equal to the number of nodes in SPOKE. This process is then repeated for each FC column in the FC matrix. Taking the dot product of the two matrices simplifies this process into one step.
We have adjusted the manuscript as follows to explain the use of the dot product:
“Then for each comparison the filtered PSEV matrix was adjusted using the FCs. This was accomplished by taking the product of a single column in the FC matrix and the filtered normalized PSEV matrix. For this step the rows (genes) of the filtered normalized PSEV matrix must be in the same order as the rows in the FC matrix. Next, each column (node) in the adjusted PSEV-matrix is summed resulting in a vector in which each element corresponds to a node in SPOKE (length = 389,297). Finally, each node is ranked as before (with the highest value in the vector ranked 389,297). In practice, this was achieved by taking the dot product of the filtered FC matrix (transposed) and the filtered normalized PSEV matrix and then ranking the resulting matrix.”
Page4 line 145-146. “Top nodes were selected using the most significant 2.5% per node type for Space vs. Ground and/or Space vs. Baseline (n=15,875; 4.1%)” can the author provide an explanation why they used the 2.5% cut of rather p-value cut-off. The significant nodes were too few or too much?
The threshold of 2.5% was picked because we are comparing two groups (thus, if the set of top nodes were distinct for each comparison then the total would be 5%). Second, we wanted to provide a manageable list of nodes for each type. If we only filtered by p-values, there would be 159,374 top nodes. Also, some nodes types would be over represented simply due to the distance from and/or relationships with the input data. Due to this, the p-value should be looked at for each node type. Of note, the total number of top nodes (14,043) is smaller than 5% because some of the top nodes are significant in both comparisons.
To follow this reviewer’s advice, we now also remove nodes with a p-value greater than 0.025 (n=120; all SideEffects) in the list that were not significant. In summary, we now filter by both top 2.5% and p-value cut-off (n=13,923; 3.6%).
Page 4 line 150: please clarify the meaning of top nodes.
The following sentence was added to the manuscript:
“Nodes ranked significantly different in either of the space travel comparisons (Space vs. Baseline and Space vs. Ground) compared to Ground vs. Baseline, were identified using the p-values from Welch’s t-test.”
Page 4 line 159 The authors wrote: “The p-values of the FCs used as input for PSEV creation were averaged for each group (Ground vs. Baseline, Space vs. Baseline, and Space vs. Ground). This gives us an estimate of how significant the gene FC was for a group as a whole.” This sentence is not correct from a statistical point of view: the average of p values is not an estimate of the significance of the gene in the whole group. Also the sentence starting at line 161 is not statistically sound (especially “that on average was more significant”). If you use arbitrary p-values combination to build a score you should state that the statistical meaning is lost. Otherwise use a p-value combination method that is statistically valid.
We agree with the reviewer. To address this we have adjusted analysis. Now we are utilizing Stouffer’s Method (similar to Fisher’s combined p-value) to combine p-values from either the Ground vs Baseline or space travel groups. Additionally, the test now reads:
“The p-values, derived when calculating the FCs used as input for PSEV creation, were combined for Ground vs. Baseline and the space travel groups (Space vs. Baseline and Space vs. Ground together) using Stouffer’s method. Each gene FC was judged on whether the average space travel group had a combined p-value that was more significant than Ground v Baseline (Supplementary Figure 1 y-axis).”
Minors
Page 5 line 199 Figure 2C does not show gene overlap. Moreover it is not clear if “these genes” refers to DEGs or to the overlap.
We have clarified the text to note that Figure 2C represents pathway overlap and that differentially expressed genes were used as an input.
Page3 Line 105 correct G in GO
Corrected.
Page3 line 124 please clarify the meaning of the numbers 389 , 297 in “PSEV is equal to the number of nodes in SPOKE (n = 389 , 297)”
This refers to the total number of nodes in SPOKE (389,297) but the confusion arises from extra spaces surrounding the comma. We have now fixed this.
Page 4 line 146: explain the meaning of 'n' and the percentage
The manuscript now reads:
Since 159,374 nodes had a p-value <0.025 in either or both space travel comparisons, top nodes were further filtered by selecting the most significant 2.5% of each node type for Space vs. Ground and/or Space vs. Baseline (n=15,801; 4.1%).
Page5 Line 184 If PMID:27965673 was meant as a reference, it is not immediately clear to me the link between the topic of the paper.
We have clarified the link to mean that liver can be considered an immune-associated organ and lymphatic endothelial cells play a role in liver injury and tumors. We have also corrected the reference to be added to the list of references and not as a PMID. (Lukacs-Kornek et al., The Role of Lymphatic Endothelial Cells in Liver Injury and Tumor Development, Front Immunol 2016)
Line 197 - 203 Figure 2 c,d caption does not match the figure and the text. Please clarify the figure caption or the text.
We have corrected the text to mean “gene sets” instead of “genes” and match the figure.
Table 1 at page 6 is not readable. Please split long content in multiple lines.
We have split the content into multiple lines and split the table into parts A and B to improve readability.
Figure 4 is not readable as well
We have made the text larger in figure 4.
Reviewer 2 Report
• Nelson et al. combined transcriptomic data from six studies from NASA GeneLab repository in combination with SPOKE to infer the phenotypic impact of biological changes during spaceflight. It is important to combine terrestrial and space data to interpret the possible connections of animal study results to observed astronaut phenotypes. This is a proof of concept approach, combining the SPOKE knowledge graph database with GeneLab data. While the paper is interesting and has potential usability for astronaut and terrestrial application of animal study findings, several clarifications and revisions must be addressed prior to publication as indicated below.
Major:
- Overall the analysis section does not present a clear easy-to-follow flow. Perhaps a flowchart delineating the various steps and how these are combined would be useful, as this is central to the manuscript as a proof-of-concept method.
- This is a reanalysis, and multiple data have been used including analyses from previous studies. However, it is very important to distinguish clearly in the writeup what has been done in this specific work and what was reused from previous studies, including appropriated citations. See Minor comments below for examples. This can be clearly labeled also if a flowchart is provided (see 1 above).
- As this is a computational approach with public datasets, for any analysis that is done in this study, I recommend that the data and code needs to be made available for reproducibility. Especially since this is a proof-of-principle for future use. All the results for "significance tests" should also be provided, including gene lists in appropriate tables (not as PDF versions). Perhaps consider providing these on Zenodo.
Minor:
- What were the selection criteria for these datasets (versus any other GeneLab datasets) - would be good to summarize the selection parameters, and why these were selected. Did the authors try other datasets too?
- For all the different software in Materials and methods please make sure to also cite the appropriate relevant manuscripts - this is lacking throughout the manuscript.
- A clarification is needed as to why there were two different versions of HTStream SeqScreener used for the data rRNA content assessments. Similarly different versions of MultiQC, and versions of STAR. It appears this is the case if the data were not reanalyzed together. Hence, if the data were obtained analyzed from the database this should be stated, and clarified in the manuscript that no new reanalysis of raw data was carried out for this study. (Materials and Methods first paragraph, lines 67-85).
- Please provide the code for running the analyses. What is the "affyNormQC.R" script, and where can it be obtained? Similarly for "annotateProbes.R" script. The "script" referencing sentence is copied directly from https://genelab-data.ndc.nasa.gov/genelab/accession/GLDS-6/ which is where the data were obtained, but this needs to be cited and not used verbatim as if this analysis was done for this manuscript. Again, please distinguish clearly which analysis was done for this manuscript, versus using the results of previous analysis.
- As the results of multiple studies are being combined, why not try to eliminate confounding effects in the integration from the beginning?
- Line 105 "G molecular function" should be "GO molecular function"
- The authors are referring to pathways, but are actually using GO terms - please clarify if GO terms or pathways are being used (lines 105-107), and Figure 2 (c).
- Use of contraction - line 131 "weren't"
- Line 128 change to "mapped to a single human gene"
- Eq. 1 should be written in mathematical notation with the symbols defined.
- In the analysis the details of "significance" are not given, nor any corrections - it is stated that p<0.05 without additional details. This occurs throughout the manuscript - please clarify all instances of "significance" with the cutoff used, and if any or no multiple hypothesis corrections were carried out.
- Table 1 is not focused in the peer review version of the manuscript.
- The authors state that "that human physiological changes observed during spaceflight can be inferred by embedding mouse gene expression data with a KG…" No human spaceflight data are actually used in this study, so these results are more putative associations. The sentence should be modified to clarify that here changes in mice are interpreted for potential human relevance.
- Are there limitations to the approach presented? Please discuss.
- How do the implementation findings compare to the findings of the original studies from which the data were obtained?
Author Response
Reviewer #2
Nelson et al. combined transcriptomic data from six studies from NASA GeneLab repository in combination with SPOKE to infer the phenotypic impact of biological changes during spaceflight. It is important to combine terrestrial and space data to interpret the possible connections of animal study results to observed astronaut phenotypes. This is a proof of concept approach, combining the SPOKE knowledge graph database with GeneLab data. While the paper is interesting and has potential usability for astronaut and terrestrial application of animal study findings, several clarifications and revisions must be addressed prior to publication as indicated below.
Major
1. Overall the analysis section does not present a clear easy-to-follow flow. Perhaps a flowchart delineating the various steps and how these are combined would be useful, as this is central to the manuscript as a proof-of-concept method.
We have added Figure 1a, where the whole process is graphically depicted
2. This is a reanalysis, and multiple data have been used including analyses from previous studies. However, it is very important to distinguish clearly in the writeup what has been done in this specific work and what was reused from previous studies, including appropriated citations. See Minor comments below for examples. This can be clearly labeled also if a flowchart is provided (see 1 above).
We have added a workflow to figure 1 and additional text to highlight that we are using data from previous studies including:
“Data representing the quantitative analysis of gene expression for each dataset was downloaded from the NASA GeneLab repository, where it had been previously analyzed and imported into R.”
3. As this is a computational approach with public datasets, for any analysis that is done in this study, I recommend that the data and code needs to be made available for reproducibility. Especially since this is a proof-of-principle for future use. All the results for "significance tests" should also be provided, including gene lists in appropriate tables (not as PDF versions). Perhaps consider providing these on Zenodo.
We deposited the code and tables on the following github repository:
https://github.com/baranzini-lab/SPOKE_NASA
Minor:
1. What were the selection criteria for these datasets (versus any other GeneLab datasets) - would be good to summarize the selection parameters, and why these were selected. Did the authors try other datasets too?
This is the first such study with a purpose of serving as a proof-of-concept. We plan to expand it by an order of magnitude to use GeneLab datasets for a more complex analysis followed by experimental validation.
The text has been modified as follows:
“The sets were selected to include multiple immune organs (thymus, spleen, immune-associated organ liver) collected from the same spaceflown mice as well as between mice flown on different missions to increase sample diversity, and to include RNA sequencing and microarray as two different sequencing methods to show that both can be used as inputs to SPOKE.”
2. For all the different software in Materials and methods please make sure to also cite the appropriate relevant manuscripts - this is lacking throughout the manuscript.
Corrected.
3. A clarification is needed as to why there were two different versions of HTStream SeqScreener used for the data rRNA content assessments. Similarly different versions of MultiQC, and versions of STAR. It appears this is the case if the data were not reanalyzed together. Hence, if the data were obtained analyzed from the database this should be stated, and clarified in the manuscript that no new reanalysis of raw data was carried out for this study. (Materials and Methods first paragraph, lines 67-85).
The reviewer is correct, the data were obtained from the database already analyzed. We have clarified it in the Materials and Methods.
4. Please provide the code for running the analyses. What is the “affyNormQC.R” script, and where can it be obtained? Similarly for “annotateProbes.R” script. The “script” referencing sentence is copied directly from https://genelab-data.ndc.nasa.gov/genelab/accession/GLDS-6/ which is where the data were obtained, but this needs to be cited and not used verbatim as if this analysis was done for this manuscript. Again, please distinguish clearly which analysis was done for this manuscript, versus using the results of previous analysis.
We have distinguished previous from new analysis and provided references for the R scripts that are found in Bioconductor packages ‘annotateTools’, etc.
5. As the results of multiple studies are being combined, why not try to eliminate confounding effects in the integration from the beginning?
Combining gene expression datasets from different studies is a challenge and no satisfactory solution yet exists to perform this task. One of the powerful advantages of SPOKE is its ability to combine results from multiple studies without having to re-analyze them as each new study is added, which is what we aimed to showcase in the manuscript.
6. Line 105 “G molecular function” should be “GO molecular function”
Corrected.
7. The authors are referring to pathways, but are actually using GO terms – please clarify if GO terms or pathways are being used (lines 105-107), and Figure 2 ©.
Clarified.
8. Use of contraction – line 131 “weren’t”
Corrected.
9. Line 128 change to “mapped to a single human gene”
Corrected.
10. Eq. 1 should be written in mathematical notation with the symbols defined.
We are now using Stouffer’s method for combining the p-vales
11. In the analysis the details of “significance” are not given, nor any corrections – it is stated that p<0.05 without additional details. This occurs throughout the manuscript – please clarify all instances of “significance” with the cutoff used, and if any or no multiple hypothesis corrections were carried out.
Throughout the manuscript we have added details to clarify “significance”. These specifically address how we select top nodes and significant genes.
12. Table 1 is not focused in the peer review version of the manuscript.
Redone to be more clearly readable.
13. The authors state that "that human physiological changes observed during spaceflight can be inferred by embedding mouse gene expression data with a KG…" No human spaceflight data are actually used in this study, so these results are more putative associations. The sentence should be modified to clarify that here changes in mice are interpreted for potential human relevance.
We have modified this sentence:
Taken together, these results show that potential human physiological changes during spaceflight can be inferred by embedding mouse gene expression data with a KG
14. Are there limitations to the approach presented? Please discuss.
The main limitation of KGs is incompleteness due to ever growing/changing nature of a KG. We have added the following to the manuscript:
KGs including SPOKE, are bounded by present day biological knowledge. As a result, inferences made through SPOKE my change as our biological data and knowledge expands.
15. How do the implementation findings compare to the findings of the original studies from which the data were obtained?
The original studies focused only on the analysis of the transcriptomic data with conventional methods (i.e. Gene ontology and pathway enrichment). However, as Reviewer 1 points out, our approach is interesting and original because it allows to connect molecular changes to phenotype changes.
We have now added the following to the discussion:
“Using a knowledge graph connecting molecular and physiological entities (among others) via biologically relevant relationships, constitutes a significant advancement for complex, heterogeneous data analysis. This approach complements conventional transcriptomics analysis by extending the biological significance to higher level phenotypes such as symptoms and side effects, which is not possible with current methods.”
Round 2
Reviewer 2 Report
The authors have made substantial improvements to the manuscript, and addressed most of the referees' comments. The following points still must be addressed:
- Please release the code on Zenodo to obtain a DOI (you can link your GitHub repository). Also, the code points to spoke v2 and other files/directories. It is unclear where these may be found - either provide these files/directories (preferable), or minimally a readme on GitHub to indicate how these may be obtained.
- As previously commented, methods should not be copied verbatim without a reference: On page 3 Lines 95-107 still describe the GeneLab microarray data processing protocol data transformation wording verbatim as written online at https://genelab-data.ndc.nasa.gov/genelab/accession/GLDS-4/ . Please cite the webpage specifically where the sentences were copied from - understandably these methods are reused, but either rephrase and/or quote. This verbatim usage should not be employed in manuscripts and must be addressed prior to publication.
Spelling: page 3 line 116: publically -> publicly
Spelling for new sentence added: page 13 line 88: my -> may.
Author Response
1. Please release the code on Zenodo to obtain a DOI (you can link your GitHub repository). Also, the code points to spoke v2 and other files/directories. It is unclear where these may be found - either provide these files/directories (preferable), or minimally a readme on GitHub to indicate how these may be obtained.
Code has now been deposited in Zenodo (DOI: 10.5281/zenodo.4408539)
2. As previously commented, methods should not be copied verbatim without a reference: On page 3 Lines 95-107 still describe the GeneLab microarray data processing protocol data transformation wording verbatim as written online at https://genelab-data.ndc.nasa.gov/genelab/accession/GLDS-4/. Please cite the webpage specifically where the sentences were copied from - understandably these methods are reused, but either rephrase and/or quote. This verbatim usage should not be employed in manuscripts and must be addressed prior to publication.
We have now referenced these methods appropriately.
Spelling: page 3 line 116: publically -> publicly
Corrected
Spelling for new sentence added: page 13 line 88: my -> may.
corrected